# Microcirculatory Disease in Patients after Heart Transplantation

**DOI:** 10.3390/jcm12113838

**Published:** 2023-06-04

**Authors:** Sylwia Iwańczyk, Patrycja Woźniak, Anna Smukowska-Gorynia, Aleksander Araszkiewicz, Alicja Nowak, Maurycy Jankowski, Aneta Konwerska, Tomasz Urbanowicz, Maciej Lesiak

**Affiliations:** 11st Department of Cardiology, Poznan University of Medical Sciences, 60-701 Poznań, Poland; patrycja.wozniak@usk.poznan.pl (P.W.); anna.smukowska-gorynia@usk.poznan.pl (A.S.-G.); aleksander.araszkiewicz@usk.poznan.pl (A.A.); alicja.nowak@usk.poznan.pl (A.N.); maciej.lesiak@skpp.edu.pl (M.L.); 2Department of Computer Science and Statistics, Poznan University of Medical Sciences, 60-701 Poznań, Poland; mjankowski@ump.edu.pl; 3Department of Histology and Embryology, Poznan University of Medical Sciences, 60-701 Poznań, Poland; akonwer@ump.edu.pl; 4Cardiac Surgery and Transplantology Department, Poznan University of Medical Sciences, 60-701 Poznań, Poland; tomasz.urbanowicz@skpp.edu.pl

**Keywords:** cardiac allograft vasculopathy, acute allograft rejection, index of microcirculatory resistance, fractional flow reserve, coronary flow reserve

## Abstract

Although the treatment and prognosis of patients after heart transplantation have significantly improved, late graft dysfunction remains a critical problem. Two main subtypes of late graft dysfunction are currently described: acute allograft rejection and cardiac allograft vasculopathy, and microvascular dysfunction appears to be the first stage of both. Studies revealed that coronary microcirculation dysfunction, assessed by invasive methods in the early post-transplant period, correlates with a higher risk of late graft dysfunction and death during long-term follow-up. The index of microcirculatory resistance, measured early after heart transplantation, might identify the patients at higher risk of acute cellular rejection and major adverse cardiovascular events. It may also allow optimization and enhancement of post-transplantation management. Moreover, cardiac allograft vasculopathy is an independent prognostic factor for transplant rejection and survival rate. The studies showed that the index of microcirculatory resistance correlates with anatomic changes and reflects the deteriorating physiology of the epicardial arteries. In conclusion, invasive assessment of the coronary microcirculation, including the measurement of the microcirculatory resistance index, is a promising approach to predict graft dysfunction, especially the acute allograft rejection subtype, during the first year after heart transplantation. However, further advanced studies are needed to fully grasp the importance of microcirculatory dysfunction in patients after heart transplantation.

## 1. Introduction

Over the last 30 years, the treatment and prognosis of patients after heart transplantation (HTx) have significantly improved, despite the older age and higher morbidity of recipients [1]. In addition, the number of transplants has increased markedly, with immunosuppressive treatment and post-transplantation care also improved. However, graft dysfunction after HTx remains a significant problem, and establishing etiology and early predictors may be crucial for patient prognosis and management.

Early graft dysfunction occurs within the first 24 h after the transplant procedure. It is classified as primary or secondary, depending on the etiology. Primary graft dysfunction (PGD) is divided into PGD-left and PGD-right ventricle, leading to ventricular dysfunction requiring pharmacological or mechanical circulatory support. PGD risk factors can be donor-related (e.g., age, cause of death, inotropic support), recipient-related (e.g., mechanical support, infection, retransplant), or surgical procedure-related (e.g., ischemia time, weight mismatch, increased blood transfusion requirement) [1]. In turn, secondary graft dysfunction (SGD) may result from hyperacute graft rejection, pulmonary hypertension, or a range of known surgical complications [2].

Late graft dysfunction includes two main subtypes: acute allograft rejection (AAR) and cardiac allograft vasculopathy (CAV). AAR-related deaths account for 11% of all deaths in the first three years following transplantation [3]. The overall prevalence of CAV increases with time. In the period from one to three years after transplantation, CAV is the leading cause of death [4]. Coronary microcirculation disease (CMD) seems to be the common factor in the pathogenesis of both AAR and CAV. Notably, microvasculature appears to be affected before any clinical manifestations of the rejection process. Furthermore, study results showed that CMD, assessed using invasive methods in the early post-transplant period, correlates with a higher risk of late graft dysfunction and death [5]. Hence, in this review, we have summarized the current knowledge on the importance of coronary microvascular disease in the pathogenesis of late graft dysfunction in HTx patients.

## 2. Coronary Microcirculation Assessment Methods

CMD is defined as a coronary microcirculation dysfunction resulting in myocardial ischemia [6,7]. Non-invasive and invasive techniques can be used for its diagnosis, with invasive assessment being the gold standard. Coronary flow reserve (CFR) and microcirculatory resistance index (IMR) are considered crucial CMD assessment parameters. According to clinical trial results, IMR ≥ 25 units or CFR < 2.0 are regarded as microcirculatory dysfunction [6,7,8,9]. The mechanisms underlying CMD include structural remodeling and functional disturbances of the coronary microcirculation [8,9,10]. As a result of vascular remodeling, the capillary wall thickens due to endothelial and smooth muscle cell hypertrophy and proliferation and perivascular fibrosis, resulting in a diameter and density decrease and a compromised coronary microvascular blood flow. Functional impairment manifests through impaired vascular reactivity to received stimuli, usually with a predominance of contraction [11,12]. The endothelium plays a crucial role in the regulation of vascular tone through the synthesis and release of endothelial-derived relaxation factors (EDRFs), including vasodilating prostaglandins (e.g., prostacyclin), nitric oxide (NO), and endothelial-dependent hyperpolarization factors (EDH), as well as endothelium-derived contractile factors (EDCFs) [13,14]. Endothelial dysfunction can be attributed to decreased production or action of EDRF or increased EDCF response, initiating a step towards atherosclerotic cardiovascular disease [15]. However, although endothelial impairment is a feature of atherosclerosis, not all types of endothelial dysfunction result from atherosclerotic buildup. There is no technique allowing direct visualization of the coronary microcirculation in vivo. The coronary microvascular function can be assessed indirectly by methods that measure myocardial blood flow (MBF) and CFR, which are usually highly dependent on the functional integrity of the coronary microcirculation. Nonetheless, the last two decades have seen the development and refinement of non-invasive cardiac imaging. Studies using positron emission tomography (PET) and cardiac magnetic resonance (CMR) for non-invasive quantification of regional MBF have shown that CMD can be detected in many clinical conditions without visible stenoses of large epicardial arteries. However, the experience using non-invasive CMD methods as predictive tools in HTX patients still needs improvement [16]. Bravo et al. reported, in a cohort of 94 transplant patients, that quantifying absolute MBF using PET improves CAV detection compared to conventional semi-quantitative myocardial perfusion imaging (MPI), providing better patient stratification [17]. The authors suggest that MBF quantification may be an essential alternative to coronary angiography for detecting CAV after HTX, eliminating the need for invasive procedures and contrast agent administration. MBF quantification using PET is an integrated functional assessment of both the epicardial coronary arteries and the microvasculature, thus enabling the collection of information about the presence of CAV at the microcirculatory level. Coronary angiography excludes obstructive coronary artery disease (CAD), and the complementary catheter-based techniques used during the procedure allow the assessment of epicardial and microvascular coronary physiology. Invasive CFR assessment measures coronary vasomotor dysfunction, including the hemodynamic effects of focal, diffuse, and small-vessel disease on myocardial perfusion. CFR is most commonly assessed using an intracoronary Doppler-tipped guidewire or thermodilution techniques and calculated as the ratio of hyperemic to rest absolute myocardial blood flow [18,19]. It should be mentioned that the limitation of the CFR is its poor reproducibility and partial dependence of the result on resting hemodynamics. Moreover, CFR cannot discriminate between epicardial and MCD (it has to be combined with FFR or iwFR for such applications) [18,19]. In turn, IMR is a novel quantitative measure of the coronary microvasculature function. It is calculated, with a combined pressure–temperature sensor-tipped coronary guidewire, as the distal coronary pressure divided by the inverse of the mean transit time during maximal hyperemia. The advantage of IMR over CFR is its high specificity for microcirculation, quantification, reproducibility, and independence from hemodynamic parameters. Importantly, an abnormal IMR was associated with worse cardiovascular disease outcomes in patients without CAD. The predictive value of IMR has also been reported in patients with acute myocardial infarction with ST-segment elevation, cardiomyopathies, and myocarditis, as well as after HTx [19]. Functional tests to assess endothelial dysfunction, manifested by pathological epicardial coronary artery or microcirculation (arterioles) spasm, include a provocation test with selective intracoronary administration of acetylcholine (ACH). The absence of the epicardial coronary artery spasm, accompanied by angina symptoms with or without concomitant ischemic ECG changes, allows for diagnosing vasoconstrictive CMD [20].

## 3. Mechanisms of Heart Transplant Rejection

AAR can be cell or antibody (humoral)—mediated and known as Acute Cellular Rejection (ACR) and Antibody-Mediated Rejection (AMR), respectively. The risk factors include younger age, the female gender of the donor or recipient, and increased HLA mismatch [21,22]. ACR is most common in the early post-HTx period, with its diagnosis established using endomyocardial biopsy (EBM). Due to the extensive follow-up cardiac biopsy protocol, most cases of ACR are diagnosed in asymptomatic patients. In a review of the Cardiac Transplant Research Database of 3367 patients with 4137 ACR episodes, only approximately 5% were diagnosed with severe hemodynamic dysfunction during a routine follow-up [14]. In a more advanced stage, left and right ventricular failure symptoms appear. It should be mentioned that improper sampling of the myocardium may result in an underestimation of the severity of rejection. Therefore, the obtained biopsy results should be considered together with clinical symptoms. Cardiac transplant biopsies are graded for ACR according to the standardized International Society for Heart and Lung Transplantation (ISHLT) nomenclature [23]: Grade 0—no rejection; Grade 1 R, mild—interstitial and/or perivascular infiltrate with up to one focus of myocyte damage; Grade 2 R, moderate—two or more foci of infiltrating with associated myocyte damage; Grade 3 R, severe—diffuse infiltrates with multifocal myocyte damage, with or without edema, hemorrhage, or vasculitis.

AMR is a less understood and more difficult-to-diagnose process, but it can potentially result in high morbidity [24,25]. Almost half of heart transplant recipients who develop rejection seven years after transplantation have evidence of AMR [26]. The diagnosis of AMR is established by histopathological evaluation of myocardial biopsies, including the following criteria [27]: (1) Damage to myocardial capillaries with an intravascular accumulation of macrophages, an intravascular thrombosis and interstitial edema, hemorrhage and neutrophilic infiltration in and around the capillaries. (2) Positive immunofluorescence or immunoperoxidase staining for AMR within capillaries, including macrophage staining of immunoglobulins (IgG, IgM, IgA), complement (C4d, C3d, C1q), and CD68. The AMR severity nomenclature was also developed by ISHLT [18] assessing endothelial activation, intravascular macrophages as well as capillary immunostaining for C4d, C3d, and CD68 macrophages: pAMR 0—negative for pathological AMR—negative both histologic and immunopathologic examinations; pAMR 1 (H+)—only histopathologic AMR—positive histologic and negative immunopathologic results; pAMR 1 (I+)—only immunopathologic AMR—negative histologic and positive immunopathologic results; pAMR 2—pathological AMR—both histologic and immunopathologic abnormalities are present; pAMR 3—severe pathological AMR—severe AMR with histopathologic signs of interstitial hemorrhage, capillary fragmentation, mixed inflammatory infiltrates, endothelial cell pyknosis, cariorexia, and severe edema.

CAV, also known as transplant coronary artery disease or cardiac transplant vasculopathy, is one of the three leading causes of death in HTx patients. The incidence of CAV concerns approximately 30% of patients five years after transplantation and 50% of patients ten years after transplantation [5]. CAV risk factors include elevated cholesterol, cytomegalovirus infection, insulin resistance, donor coronary heart disease, younger recipient, and history of acute rejection. Other causes of allograft failure include myocardial diseases such as amyloidosis, sarcoidosis, giant cell myocarditis, hereditary hemochromatosis, and malignancies such as primary heart lymphoma [28]. CAV typically affects all epicardial allograft arteries and is characterized by diffuse concentric intimal hypertrophy and vascular remodeling [29,30]—similar changes concern coronary microcirculation. Intracoronary ultrasonography (IVUS) shows that intima hyperplasia mainly occurs in the first year after transplantation [31]. ISHLT CAV classification accurately and independently predicts long-term prognosis after HTx. CAV can be classified as absent (CAV 0), mild (CAV 1), moderate (CAV 2), or severe (CAV 3) [1].

## 4. Coronary Microcirculation Disease in Patients after Heart Transplantation

Crea et al. showed that the patients with ACR had significantly higher IMR values at one month compared to those without ACR in observation (23.1 ± 8.6 versus 16.8 ± 11.1, *p* = 0.002) [10]. In addition, IMR was associated with the higher risk of ACR (adjusted hazard ratio [aHR], 1.18 [95% confidence interval [CI], 1.04–1.34], *p* = 0.011), and the optimal cutoff value of IMR to predict ACR was 15 [22]. The authors revealed that adding IMR to clinical variables significantly increased the ability to predict and reclassify the risk of ACR. Moreover, microvascular dysfunction was correlated with a higher probability of death, graft failure, or allograft vasculopathy five years after HTx (hazard ratio, 2.52 [95% CI, 1.04–5.91]). The increased IMR values in the first year after HTx were also associated with worse graft function and simultaneously less promising clinical outcomes [32]. Haddad et al. evaluated 63 HTx recipients for coronary physiology at one-year post-transplant follow-up. Microcirculatory dysfunction, defined as IMR > 20, was observed in 46% of patients after one year. A history of AAR and undersized donor hearts were associated with microvascular dysfunction at one year, with an odds ratio of 4.0 (1.3–12.8) and 3.6 (1.2–11.1), respectively. Additionally, patients with microvascular dysfunction had lower cardiac index (3.1 ± 0.7 versus 3.5 ± 0.7 L/min per m2; *p* = 0.02) and mild graft dysfunction assessed with the use of echocardiography-derived left and right myocardial performance indices ([0.54 ± 0.09 versus 0.43 ± 0.09; *p* < 0.01] and [0.47 ± 0.14 versus 0.32 ± 0.05; *p* < 0.01], respectively). The authors unanimously concluded that microvascular dysfunction correlated with a higher probability of death, graft failure, or allograft vasculopathy five years after HTx (hazard ratio, 2.52 [95% CI, 1.04–5.91]). Increased IMR values in the first year after HTx were also associated with worse graft function and, simultaneously, less promising clinical outcomes [26]. Another study published by Ahn et al. enrolled 237 patients from five international cohorts [32]. IMR was measured at a median of seven weeks after HTx. The primary outcome was the incidence of AAR during the first year post-transplantation, while the secondary outcomes comprised major adverse cardiovascular events (MACE, i.e., the composite of death, retransplantation, myocardial infarction, stroke, graft dysfunction, and hospital readmission) within ten years. After the thorough analyses, the authors concluded that during the first year, IMR values correlated proportionally with a higher risk of AAR (aHR: 1.04; 95% CI: 1.02–1.06; *p* < 0.001). In addition, the incidence of AAR in patients with an IMR ≥ 18 was 23.8%, whereas the incidence of AAR in those with an IMR < 18 was 6.3% (aHR: 3.93; 95% CI: 1.77–8.73; *p* = 0.001). MACE occurred in 86 (36.3%) patients within a 10-year period, and its risk was significantly higher with IMR ≥ 18 (aHR: 1.02; 95% CI: 1.01–1.04; *p* = 0.005). Considering the above, measuring the IMR early after HTx might identify patients at higher risk of ACR and MACE and optimize and improve post-transplant management. In addition, IMR has proven valuable in assessing the treatment response [23]. Yang et al. concluded that patients with decreased or stable IMR (from baseline to one year) presented a longer period without MACE compared to patients with an increased IMR (66% versus 36%; *p* = 0.03) [33]. Moreover, the authors selected IMR ≥ 20 at one year (HR, 3.93; 95% CI, 1.08–14.27; *p* = 0.04) as an independent predictor of death or retransplantation. Okada et al. [34] measured CFR and IMR within eight weeks post-transplantation in another study. In the first year of follow-up, 25.0% of patients experienced AAR. They presented higher IMR and lower CFR at baseline compared with those without AAR. In the multivariate analysis, only baseline IMR ≥ 16.0 was independently associated with AAR during the first year, demonstrating a high negative predictive value (96.7%). Moreover, in the meta-analysis, including 616 patients from five trials, the authors assessed the predictive potential of IMR for subsequent AAR and long-time cardiac events [35]. Both the risk of AAR (RR: 4.08; 95% CI: 2.69–6.17; *p* < 0.001) and long-term cardiac events (RR: 2.14; 95% CI: 1.44–3.19; *p* < 0.001) was significantly higher in IMR-high compared to IMR-low group. Patients treated with HTX in the high IMR group had better predictive efficacy than the low IMR group.

As of this day, the data on the correlation between AMR and IMR are limited. So far, single reports have indicated a potential relationship between microcirculatory dysfunction and AMR. In their study, Afzali et al. assessed molecular diagnostics in human paraffin-embedded EMBs with AMR, classified according to ISHLT criteria. The gene set expression was subsequently compared between ISHLT diagnoses and correlated with donor-specific antibodies, endothelial injury by electron microscopy, and prognosis. In AMR patients (n = 17) with sequential biopsies, upregulated gene set expression was associated with a worse prognosis (*p* = 0.034). These findings were validated in an independent set of EBMs. The main conclusion was that the biopsy-based molecular examination of antibody-mediated microcirculation injury could improve the diagnosis of AMR in patients after HTx [36].

CAV is an independent prognostic factor for allograft rejection and survival rate. The origin of CAV is multifactorial, including immune and non-immune factors. The abnormal immune processes appear to be of paramount importance, as allograft vasculopathy develops in the donor’s but not in the recipient’s arteries. CAV probably reflects an accelerated “normal” healing process following transplant-induced immune damage [30]. Endothelial injury and dysfunction are prevalent early after transplant, possibly as precursors to overt CAV [37]. Hollenberg et al. assessed endothelial dysfunction in seventy-three patients yearly, beginning at transplantation [38]. After seven years of the study, the endpoint, defined as angiographic evidence of CAV (>50% stenosis) or cardiac death (graft failure or sudden death), occurred in 14 patients (6 CAV and eight deaths). The patients exhibited reduced epicardial (11.1 ± 2.9% stenosis vs. 1.7 ± 2.2% dilation, *p* = 0.01) and microvascular (75 ± 20% increase in flow vs. 149 ± 16%, *p* = 0.03) endothelium-dependent response to ACH, compared to patients who did not reach the endpoint. In conclusion, endothelial dysfunction, as reflected by abnormal ACH responses, preceded clinical endpoint development. Moreover, studies report that microcirculatory dysfunction can predict CAV and, when assessed regularly, can prevent graft dysfunction’s progression and improve patients’ survival after HTx [39]. Lee et al. emphasized the importance of IMR evaluation early after HTx to identify patients at risk for developing CAV [40]. In the study of Faron et al. FFR significantly worsened in the one-year follow-up after HTx (0.90 ± 0.05 at baseline to 0.85 ± 0.06 after one year, *p* = 0.004), while IMR significantly improved (29.2 ± 15.9 at baseline to 19.3 ± 7.6 units at one year, *p* = 0.007). Both FFR and IMR correlated with plaque volume measured using IVUS [15]. In their study, Chih et al. assessed the risk of CAV development and progression in the first year after heart transplantation in 82 patients and evaluated the predictive value of early assessment of atherosclerotic plaque morphology and coronary microcirculation [41]. They showed that fibrotic plaque on OCT and IMR early posttransplant predicts CAV progression in the first year of transplantation.

Limitations of invasive assessment of microcirculation in heart transplant recipients should also be discussed. One is the risk of an atrioventricular block (AVB) in a denervated heart during intravenous infusion or intracoronary administration of adenosine. Studies have shown that adenosine induces AVB in healthy pediatric and young adult heart transplant recipients with minimal risk when low initial doses are used and therapy gradually escalates [42]. Using regadenoson, a selective A2A adenosine receptor agonist, instead of adenosine, eliminates the risk of AVB. However, the results of the use of regadenoson in this patient population have yet to be performed. In addition, invasive MVD diagnostics slightly prolong the coronary angiography procedure. However, due to the diffuse pattern of microvascular damage, evaluation in all vessels is not required. The assessment of coronary microcirculation is performed only in the artery supplying the largest area of the myocardium, most often the anterior descending artery (LAD).

The studies discussed above were mainly aimed at demonstrating the importance of early assessment of coronary microcirculation in patients after heart transplantation in predicting long-term complications, including AAR and CAV. The question remains about the clinical significance of the obtained results and how to use them to improve the prognosis of post-transplant patients in long-term follow-up. Optimal immunosuppressive therapy prevents rejection and damage to the donor organ while minimizing complications due to immunosuppressive treatment [43]. Sub-therapeutic serum levels of immunosuppressants are a significant factor in developing rejection after transplantation. Inadequate maintenance immunosuppression may be due to patient noncompliance with medications, nonadherence due to medication side effects or cost, inadequate medication absorption, drug–drug interactions, or infection [44]. The selection of patients with an increased risk of rejection and CAV at the preclinical stage based on the parameters of coronary microcirculation dysfunction will prompt the clinician to conduct a detailed analysis of immunosuppressive therapy in selected patients. In addition, an individual follow-up plan for high-risk patients also seems warranted. Modification of immunosuppressive treatment in patients with CMD, even before the onset of rejection symptoms, remains highly interesting. However, in such a management strategy, one should remember the increased risk of side effects of immunosuppressive drugs. Undoubtedly, further research is needed to assess the benefits and risks associated with the individualization of treatment of patients after heart transplantation based on the assessment of coronary microcirculation dysfunction.

The relationship between coronary microcirculation and late graft dysfunction is summarized in Figure 1.

## 5. Conclusions

The last decade contributed to significant advancement in methods of microcirculatory evaluation. An invasive assessment of the coronary microcirculation, including the IMR measurement, is the most promising approach to predict subsequent graft dysfunction (especially the AAR subtype) during the first year after HTx. However, AAR is still a poorly understood and difficult-to-predict process that can result in high morbidity [45]. Therefore, further advanced studies are needed on the importance of microcirculatory dysfunction in patients after HTx.

## Figures and Tables

**Figure 1 jcm-12-03838-f001:**
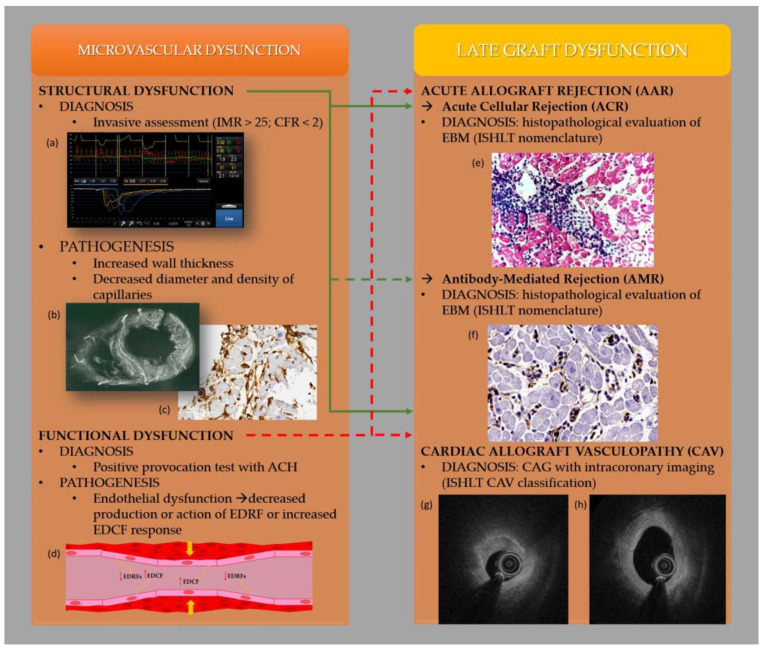
Relationship between coronary microcirculation and late graft dysfunction. (**a**) The result of the invasive assessment of the coronary microcirculation; (**b**) The results of immunohistochemical stainings of representative myocardial biopsy sample. The paraffin-embedded, microtome-cut slices of myocardial tissue were stained using a monoclonal anti-CD34 antibody, in a 1:25 dilution, to visualize the microvasculature (dark brown); (**c**) The figure shows the extensive network of coronary microcirculation vessels in a post mortem angiographic examination; (**d**) The figure presents endothelium dysfunction in the coronary microcirculation vessel, which leads to decreased production or action of endothelial-derived relaxation factors (EDRF) or increased endothelium-derived contractile factors (EDCFs), and consequently vasoconstriction; (**e**) Endomyocardial biopsy presented ACR, Grade 2 R, moderate—two or more foci of infiltrating with associated myocyte damage; (**f**) Endomyocardial biopsy presented AMR with macrophage staining of CD68; (**g**) Optical coherence tomography (OCT)—cross section of a coronary artery of a patient diagnosed with CAV. Vasculopathy significantly narrowing the lumen of the vessel; (**h**) Optical coherence tomography (OCT)—cross section of a coronary artery of a patient diagnosed with CAV. Significant thickening of the neointima without narrowing the lumen of the artery.Studies revealed an association between structural coronary microvascular dysfunction, invasively measured by the index of microcirculatory resistance (IMR) and coronary flow reserve (CFR), with both Acute Cellular Rejection (ACR) and Cardiac Allograft Vasculopathy (CAV) (solid green line). The relationship between structural microcirculatory dysfunction and Antibody-Mediated Rejection (AMR) remains poorly established (green dotted line). Similarly, a direct connection between functional coronary microvascular dysfunction and Acute Allograft Rejection (AAR) or CAV has not been demonstrated. However, the author points to this relationship due to the importance of endothelial dysfunction in both processes (red dotted line). (**a**) The result of the invasive assessment of the coronary microcirculation. (**b**) The results of immunohistochemical stainings of representative myocardial biopsy sample. The paraffin-embedded, microtome-cut slices of myocardial tissue were stained.

## Data Availability

Not applicable.

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
