# Peer review of "Microcirculatory Disease in Patients after Heart Transplantation"

_jcm, 2023, doi:10.3390/jcm12113838_

Round 1

Reviewer 1 Report

In this review the auhors summarize the evidence relating cardiac allograft dysfunction in the form of rejection and artery vasculopathy and microcirculatory dysfunction. The paper is well-written and clinically relevant. However, as non-invasive means tend to substitute invasive ones, more emphasis on non-invasive evaluation of microcirculation  may be advisable. Are there any data on non-invasive markers of microvascular dysfunction in these patients (for example echocardiography-based coronary flow velocity reserve, PET, MFR-SPECT)? If yes, they should be included. If no, a relevant statement is also needed. Minor corrections are needed.

line 36: patient prognosis and management.

line 48: incidence or prevalence?

line 66: functional dysfunction is an exaggeration. Vascular dysfunction or functional impairment better.

lines 71-76: endothelial impairment is a feature of atherosclerosis, but not all types of endothelial dysfunction are due to atherosclerosis. This should be made clear.

line 77: at present, no anatomical imaging method (either invasive or non-invasive) is clinically available for MCD, thus only functional studies are applied.

lines 86-87: reference needed. Moreover, CFR cannot discriminate between epicardial and MCD (it has to be combimed with FFR or iwFR for such applications).

Acceptable english use, only minor corrections needed.thus worth publication

Author Response

Dear Editor, Dear Reviewer

We are grateful for peer-reviewing and recommendations concerning our manuscript entitled: “The microcirculatory disease in patients after heart transplantation.” We appreciate the Reviewer’s effort very much and hope that the answers contained below will be satisfactory. The article has undergone professional language proofreading, as proof of which we attach a certificate.

Yours faithfully,
Sylwia Iwańczyk

Reviewer 1

  1. In this review the authors summarize the evidence relating cardiac allograft dysfunction in the form of rejection and artery vasculopathy and microcirculatory dysfunction. The paper is well-written and clinically relevant, thus worth publication. However, as non-invasive means tend to substitute invasive ones, more emphasis on non-invasive evaluation of microcirculation  may be advisable. Are there any data on non-invasive markers of microvascular dysfunction in these patients (for example echocardiography-based coronary flow velocity reserve, PET, MFR-SPECT)? If yes, they should be included. If no, a relevant statement is also needed. Minor corrections are needed.

Response: Thank you very much for your valuable comments. We added the part about non-invasive methods.

“There is no technique allowing direct visualization of the coronary microcirculation in vivo. The coronary microvascular function can be assessed indirectly by methods that measure myocardial blood flow (MBF) and CFR, which are usually highly dependent on the functional integrity of the coronary microcirculation. Nonetheless, the last two decades have seen the development and refinement of non-invasive cardiac imaging. Studies using positron emission tomography (PET) and cardiac magnetic resonance (CMR) for non-invasive quantification of regional MBF have shown that CMD can be detected in many clinical conditions without visible stenoses of large epicardial arteries. However, the experience using non-invasive CMD methods as predictive tools in HTX patients still needs improvement [15]. Bravo et al. reported in a cohort of 94 transplant patients that quantifying absolute MBF using PET improves CAV detection compared to conventional semi-quantitative myocardial perfusion imaging (MPI), providing better patient stratification [16]. The authors suggest that MBF quantification may be an essential alternative to coronary angiography for detecting CAV after HTX, eliminating the need for invasive procedures and contrast agent administration. MBF quantification using PET is an integrated functional assessment of both the epicardial coronary arteries and the microvasculature, thus enabling the collection of information about the presence of CAV at the microcirculatory level.”

  1. line 36: patient prognosis and management.

Response: Thank you, we corrected it.

  1. line 48: incidence or prevalence?

Response: Thank you, we corrected it.

  1. line 66: functional dysfunction is an exaggeration. Vascular dysfunction or functional impairment better.

Response: Thank you, we corrected it.

  1. lines 71-76: endothelial impairment is a feature of atherosclerosis, but not all types of endothelial dysfunction are due to atherosclerosis. This should be made clear.

Response: Thank you, we have clarified as you suggested.

  1. line 77: at present, no anatomical imaging method (either invasive or non-invasive) is clinically available for MCD, thus only functional studies are applied.

Response: Thank you, we changed the whole paragraph as above.

  1. lines 86-87: reference needed. Moreover, CFR cannot discriminate between epicardial and MCD (it has to be combined with FFR or iwFR for such applications).

Response: Thank you, we have completed the references and sentence.  

Reviewer 2 Report

I am grateful for the opportunity to review the manuscript of Iwańczyk et al. "The microcirculatory disease in patients after heart transplantation". In this review, the authors consider the problem of microvascular disease in patients after heart transplantation, methods for its diagnosis, and the impact on prognosis. The authors summarized the available literature on this issue and wrote a review that may be useful to both researchers and practitioners dealing with patients after heart transplantation. However, when reviewing, I had questions and comments to which I would like to receive answers from the authors.

1. In a scientific review, I would like to see the most recent information, however, among the references, 18 sources are older than 5 years and 8 are older than 10 years. It is clear that the topic of the review is rather narrow, but it seems that there are recent publications on most of the topics covered. I can cite the article by Chih et al (1) as an example.

2. It is also surprising that the authors did not consider in their review one of the few meta-analyses on the prognostic value of microcirculation in patients with heart transplantation (2). For example, the main results of this meta-analysis ("A total of 616 patients were studied in five trials. There were significant differences in subsequent AAR (RR = 4.08; 95% CI: 2.69~6.17; P = 0.000) or long-time cardiac events (RR=2.14; 95% CI: 1.44~3.19; P=0.000) between IMR-high and IMR-low patients in the forest plots. group") would completely confirmed the authors' conclusions.

3. I understand that this may be somewhat beyond the scope of the review, but I would like a more detailed clinical significance of this review. For example, how can treatment tactics change when microvascular dysfunction is detected in patients after heart transplantation?

4. Minor:

The bibliography contains a lot of redundant information in references to journal articles ("[Internet] [cited 2023 Mar 10] Available from" are redundant words, they are only necessary for references to Internet sources).

On the contrary, references to books (11, 25) contain insufficient bibliographic information. Probably it should be more complete (3 and 4).

References:

1.     Chih S, Chong AY, Džavík V, So DY, Aleksova N, Wells GA, Bernick J, Overgaard CB, Stadnick E, Mielniczuk LM, Beanlands RSB, Ross HJ. Fibrotic Plaque and Microvascular Dysfunction Predict Early Cardiac Allograft Vasculopathy Progression After Heart Transplantation: The Early Post Transplant Cardiac Allograft Vasculopathy Study. Circ Heart Fail. 2023 May 11:e010173. doi: 10.1161/CIRCHEARTFAILURE.122.010173.

2.     Lu Z, Song G, Bai X. Predictive Efficacy of the Index of Microcirculatory Resistance for Acute Allograft Rejection and Cardiac Events After Heart Transplantation: A Systematic Review and Meta-Analysis. Heart Surg Forum. 2022 Nov 30;25(5):E784-E792. doi: 10.1532/hsf.4899.

3.     Shimokawa H. Coronary Vasomotion Abnormalities. Springer Nature, 2020; 155

Ludhwani D, Abraham J, Kanmanthareddy A. Heart Transplantation Rejection. 2022 Sep 19. In: StatPearls [Internet]. Treasure Island (FL): StatPearls Publishing; 2023 Jan–

No comments

Author Response

Dear Editor, Dear Reviewer

We are grateful for peer-reviewing and recommendations concerning our manuscript entitled: “The microcirculatory disease in patients after heart transplantation.” We appreciate the Reviewer’s effort very much and hope that the answers contained below will be satisfactory. The article has undergone professional language proofreading, as proof of which we attach a certificate.

Yours faithfully,
Sylwia Iwańczyk

I am grateful for the opportunity to review the manuscript of Iwańczyk et al. "The microcirculatory disease in patients after heart transplantation". In this review, the authors consider the problem of microvascular disease in patients after heart transplantation, methods for its diagnosis, and the impact on prognosis. The authors summarized the available literature on this issue and wrote a review that may be useful to both researchers and practitioners dealing with patients after heart transplantation. However, when reviewing, I had questions and comments to which I would like to receive answers from the authors.

  1. In a scientific review, I would like to see the most recent information, however, among the references, 18 sources are older than 5 years and 8 are older than 10 years. It is clear that the topic of the review is rather narrow, but it seems that there are recent publications on most of the topics covered. I can cite the article by Chih et al (1) as an example.

Response: Thank you for your comment. We added the suggested publications to the review.

“In their study, Chih et al. assessed the risk of CAV development and progression in the first year after heart transplantation in 82 patients, and evaluated the predictive value of early assessment of atherosclerotic plaque morphology and coronary microcirculation [40]. They showed that fibrotic plaque on OCT and IMR early posttransplant predicts CAV progression in the first year of transplantation.”

  1. It is also surprising that the authors did not consider in their review one of the few meta-analyses on the prognostic value of microcirculation in patients with heart transplantation (2). For example, the main results of this meta-analysis ("A total of 616 patients were studied in five trials. There were significant differences in subsequent AAR (RR = 4.08; 95% CI: 2.69~6.17; P = 0.000) or long-time cardiac events (RR=2.14; 95% CI: 1.44~3.19; P=0.000) between IMR-high and IMR-low patients in the forest plots. group") would completely confirmed the authors' conclusions.

Response: Thank you for your comment. We added the suggested publications to the review.

“Moreover, in the meta-analysis, including 616 patients from five trials, the authors assessed the predictive potential of IMR for subsequent AAR and long-time cardiac events [34]. Both the risk of AAR (RR: 4.08; 95% CI: 2.69-6.17; p<0.001) and long-term cardiac events (RR: 2.14; 95% CI: 1.44-3.19; p<0.001) was significantly higher in IMR-high compared to IMR-low group. Patients treated with HTX in the high IMR group had better predictive efficacy than the low IMR group.”

  1. I understand that this may be somewhat beyond the scope of the review, but I would like a more detailed clinical significance of this review. For example, how can treatment tactics change when microvascular dysfunction is detected in patients after heart transplantation?

Response: Thank you for your suggestion. We added the additional paragraph.

“The studies discussed above were mainly aimed at demonstrating the importance of early assessment of coronary microcirculation in patients after heart transplantation in predicting of long-term complications, including AAR and CAV. The question remains about the clinical significance of the obtained results and how to use them to improve the prognosis of post-transplant patients in long-term follow-up. Optimal immunosuppressive therapy prevents rejection and damage to the donor organ while minimizing complications due to immunosuppressive treatment [43]. Subtherapeutic through levels is a significant factor in the development of posttransplant rejection. Inadequate maintenance immunosuppression may be due to patient noncompliance with medications, nonadherence due to medication side effects or cost, inadequate medication absorption, drug–drug interactions, or infection [44]. The selection of patients with an increased risk of rejection and CAV at the preclinical stage based on the parameters of coronary microcirculation dysfunction will prompt the clinician to conduct a detailed analysis of immunosuppressive therapy in selected patients. In addition, an individual follow-up plan for high-risk patients also seems warranted. Modification of immunosuppressive treatment in patients with CMD, even before the onset of rejection symptoms, remains highly interesting. However, in such a management strategy, one should remember the increased risk of side effects of immunosuppressive drugs. Undoubtedly, further research is needed to assess the benefits and risks associated with the individualization of treatment of patients after heart transplantation based on the assessment of coronary microcirculation dysfunction.”

  1. Minor:

The bibliography contains a lot of redundant information in references to journal articles ("[Internet] [cited 2023 Mar 10] Available from" are redundant words, they are only necessary for references to Internet sources).

On the contrary, references to books (11, 25) contain insufficient bibliographic information. Probably it should be more complete (3 and 4).

References:

  1. Chih S, Chong AY, Džavík V, So DY, Aleksova N, Wells GA, Bernick J, Overgaard CB, Stadnick E, Mielniczuk LM, Beanlands RSB, Ross HJ. Fibrotic Plaque and Microvascular Dysfunction Predict Early Cardiac Allograft Vasculopathy Progression After Heart Transplantation: The Early Post Transplant Cardiac Allograft Vasculopathy Study. Circ Heart Fail. 2023 May 11:e010173. doi: 10.1161/CIRCHEARTFAILURE.122.010173.
  2. Lu Z, Song G, Bai X. Predictive Efficacy of the Index of Microcirculatory Resistance for Acute Allograft Rejection and Cardiac Events After Heart Transplantation: A Systematic Review and Meta-Analysis. Heart Surg Forum. 2022 Nov 30;25(5):E784-E792. doi: 10.1532/hsf.4899.
  3. Shimokawa H. Coronary Vasomotion Abnormalities. Springer Nature, 2020; 155
  4. Ludhwani D, Abraham J, Kanmanthareddy A. Heart Transplantation Rejection. 2022 Sep 19. In: StatPearls [Internet]. Treasure Island (FL): StatPearls Publishing; 2023 Jan–

Response: Thank for your comment. We corrected all references according above suggestions. 

Round 2

Reviewer 2 Report

The authors did a great job of correcting the manuscript, responded to all my comments and took them into account when correcting. I have no other comments.

No comments